# Factors Related to Quality of Life of Hemodialysis Patients during the COVID-19 Pandemic

**DOI:** 10.3390/healthcare11081155

**Published:** 2023-04-17

**Authors:** Li-Yun Szu, Chih-Hsiang Chang, Suh-Ing Hsieh, Whei-Mei Shih, Lan-Mei Huang, Mei-Chu Tsai, Su-Mei Tseng

**Affiliations:** 1Department of Nursing, Taoyuan Chang Gung Memorial Hospital, Taoyuan City 33372, Taiwan; szu4069@cgmh.org.tw (L.-Y.S.);; 2Department of Nursing, Chang Gung University of Science and Technology, Taoyuan City 33302, Taiwan; 3Kidney Research Center, Department of Nephrology, Chang Gung Memorial Hospital; Graduate Institute of Clinical Medical Sciences, College of Medicine, Chang Gung University, Taoyuan City 33302, Taiwan; 4Graduate Institute of Health Care, Chang Gung University of Science and Technology, Taoyuan City 33302, Taiwan; 5Hemodialysis Unit, Taoyuan Chang Gung Memorial Hospital, Taoyuan City 33372, Taiwan

**Keywords:** end-stage renal disease, hemodialysis, self-participation, perceptions of hemodialysis, quality of life, COVID-19 pandemic

## Abstract

Taiwan had the second highest number globally of end-stage renal disease patients undergoing treatment in 2018. A meta-analysis of Chen et al. (2021) showed the incidence and mortality rates of COVID-19 were 7.7% and 22.4%, respectively. Few studies have explored the effects of patients’ self-participation and perceptions of hemodialysis on their quality of life. This study aimed to explore the factors related to hemodialysis patients’ quality of life during the COVID-19 pandemic. This study was a descriptive correlational study. Patients were recruited (n = 298) from the hemodialysis unit of a medical center in northern Taiwan. Variables included patients’ sociodemographic, psychological, spiritual, and clinical characteristics (i.e., perceived health level, comorbidities, hemodialysis duration, weekly frequency, transportation, and accompaniment during hemodialysis), perceptions of hemodialysis, self-participation in hemodialysis, and health-related quality of life (KDQOL-36 scale). Data were analyzed using descriptive and bivariate and multivariate linear regression. Multivariate linear regression, after adjusting for covariates, showed that anxiety, self-perceived health status, two vs. four comorbidities, and self-participation in hemodialysis were significantly associated with quality of life. The overall model was significant and accounted for 52.2% (*R*^2^ = 0.522) of the variance in quality of life during hemodialysis (adjusted *R*^2^ = 0.480). In conclusion, the quality of life of hemodialysis patients with mild, moderate, or severe anxiety was poorer, whereas that of patients with fewer comorbidities, higher self-perceived health status, and higher self-participation in hemodialysis was better.

## 1. Introduction

According to the United States Renal Data System [1] Annual Data Report, the prevalence of end-stage renal disease (ESRD) increased from 2.6% in 2010 to 3.9% in 2018. As of late 2018, the proportion of people on hemodialysis (HD) was 86.2%. In 2018, Mexico had the highest incidence of ESRD worldwide at 594 per million people (pmp), followed by Taiwan at 523 pmp. Taiwan had the highest prevalence of dialysis in the general population at 3429 pmp, followed by Japan at 2591 pmp. All-cause mortality of patients on dialysis showed a 16–37% increase in 2020 compared to 2017–2019 [2].

Hemodialysis is the priority procedure for the rapid lifesaving of patients with end-stage renal disease. However, this treatment cannot completely replace functioning kidneys, so many physical symptoms of discomfort occur. The most common symptom is fatigue, while dialysis disequilibrium syndrome is a frequent symptom at the beginning of hemodialysis [3]. Other symptoms include dialysis hypotension, high blood pressure, chest pain, palpitations, dyspnea, restless legs, nausea, vomiting, malnutrition, dry mouth, arteriovenous shunt bleeding, skin itching, and electrolyte imbalance [4,5,6,7]. In addition, the exhaustion after 3–4 h of hemodialysis two to three times per week makes it impossible to lead a normal life and causes role restrictions or social isolation [8,9]. The constant visits to various outpatient clinics and repeated hospitalizations also experienced by patients may lead to a feeling of helplessness, which may in turn affect their quality of life. In addition, hemodialysis patients have to deal with the coexistence of physical symptoms and a life of dialysis. They tend to passively accept health and education information, which may not meet their individual health needs, often causing them to feel powerless and as if they are losing control over their lives and diseases [10]. Tsay and Healstead (2002) suggested that authoritative health education programs are ineffective for changing patients’ lifestyles or self-care behaviors [11].

Self-participation is an important patient-centered nursing strategy that can promote patients’ health empowerment. However, most previous studies have focused on health empowerment and self-management strategies, and there is continuing emphasis on the professional perspective in nursing theory [12]. Few studies have focused on self-participation from the patient’s point of view [13]. How to start the process of patient self-participation and what are patients’ subjective experiences of self-participation are important questions. Therefore, understanding the subjective perception of how Taiwanese hemodialysis patients live a life of dialysis and more comprehensively exploring the concept of self-participation are crucial.

Hemodialysis patients are often at high risk of exposure to COVID-19 during their HD sessions [14,15,16,17]. The accelerated immunosenescence, the age-related decrease in immune functions and inflammaging, and the low-grade upregulation of certain pro-inflammatory responses caused chronic activation and dysfunction of the innate immune system are relevant with chronic kidney disease [18]. HD patients have an extremely short time between symptom onset and death that may be caused by lack of appropriate infection control in the early phase of the COVID-19 infectious disease [18]. Chen et al. [14,15] systematic review and meta-analysis showed that the COVID-19 incidence and overall mortality among HD patients (n = 396,062) were 7.7% and 22.4%, respectively. They are also concerned about being infected or spreading the disease to their family members, which makes them feel stigmatized. The uncertainties faced by HD patients during the COVID-19 pandemic include the fear and anxiety caused by epidemic control policies, pressure from healthcare professionals, and the shortage of personal protective equipment. Although healthcare professionals have emphasized preventing COVID-19 transmission through dialysis devices to safeguard patients’ physiological functioning, their psychological aspects remain neglected [19].

Due to different methodologies, the literature shows inconsistent results for some factors related to sociodemographic [20,21,22,23,24,25,26,27,28,29,30,31,32,33,34,35], psychological [23,34,36,37,38,39], and spiritual [23,40,41,42,43] aspects, and clinical characteristics [26,27,28,29,30,31,32,33] of hemodialysis patients. However, religion [23,39,40,41,42,43], social support [44,45], anxiety, and depression during COVID-19 [46,47], spiritual well-being [23,39,40,41,42,43], perceived health status [34,48], transportation to and from hemodialysis [49], the number of hemodialysis sessions per week [29,30], and feelings about hemodialysis and self-participation [21,28,50,51,52] have been less explored, and accompaniment during hemodialysis sessions lacks discussion. Therefore, this study aimed to explore factors related to the quality of life of hemodialysis patients during the COVID-19 pandemic. It is hoped that the findings of this study can be used as a reference for medical staff to provide holistic care for hemodialysis patients.

## 2. Materials and Methods

### 2.1. Design

This study involved a descriptive correlational design in which questionnaires were used to conduct a cross-sectional survey.

### 2.2. Study Framework

The independent variables of this correlational study framework were sociodemographic, mental and spiritual, and clinical characteristics, perceptions of HD, and self-participation in HD, while the dependent variable was quality of life for determining the relationship between the independent variables and the dependent variable (Figure 1).

### 2.3. Participants and Setting

The participants were recruited via purposive sampling from the target population (*n* = around 750) at the outpatient HD clinic of a teaching hospital in northern Taiwan. The inclusion criteria for patients were regular HD for more than one year for physician-diagnosed ESRD, free of COVID-19 disease, conscious, aged >20 years, able to communicate in Taiwan Mandarin or Taiwanese, normal hearing or use of hearing aids for hearing assistance, and agreement to participate in this study. The following HD patients were excluded from the study: those claiming to be in an unstable condition of their illness (e.g., cancer or severe heart, lung, liver, or kidney dysfunction), critically ill patients, and patients with mental disorders. The sample size for this correlational study was estimated using multiple linear regression (fixed model; *R*^2^ deviation from zero) in G*power v.3.1 software (Heinrich-Heine-University, Düsseldorf, German), with alpha set to 0.05, power set to 0.8, effect size (f^2^) set to 0.15, and 42 predictors. The estimated sample size was 219. Concerning the number of valid questionnaires and factor analysis for exploring construct validity of the new designated scale (perceptions of and self-participation), the actual sample size was larger than the estimated sample size.

### 2.4. Instruments

The study instruments included scales for measuring patients’ (1) sociodemographic characteristics; (2) psychological and spiritual characteristics; (3) clinical characteristics; (4) HD perception and degree of self-participation in a life of HD; and (5) Kidney Disease Quality of Life 36-item short-form survey (KDQOL-36; scale-abridged version 1.3). The instruments were Chinese versions and permission was obtained for their use from their Chinese developers. Institutional Review Board approval was also sought for their application (Appendix A).

#### 2.4.1. Sociodemographic Characteristics

The sociodemographic characteristics included in the study were age, sex, marital status, education level, religion, employment status, and social support. The Social Support Scale for HD patients developed by Lin et al. [45] explores the social support provided to HD patients by family members, friends, and healthcare workers through 17 items on 2 subscales: support from family and friends, and support from healthcare workers. The items were measured on a scale of 1 to 4 points: 1 indicating “never”; and 4 indicating “all the time.” Two items in the support from family and friends subscale were reverse-coded. A higher score indicated a stronger support system for the patients. The internal consistency (Cronbach’s α) of the total scale was 0.76, while that of the support from family and friends and support from healthcare workers subscales was 0.81 and 0.50, respectively [52]. In this study, Cronbach’s α of the total scale of the sample was 0.92, while that of the support from family and friends and support from healthcare workers subscales was 0.77 and 0.95, respectively.

#### 2.4.2. Psychological and Spiritual Characteristics

The 21-item Beck Anxiety Inventory (BAI) measures the degree to which physical anxiety symptoms have disrupted a patient’s life within the previous month. The severity of disruption is measured on a scale of 0 to 3 points: 0 indicates no impact, and 3 indicates severe impact. A higher overall score indicates more severe anxiety symptoms: 0–7 indicates minimal symptoms, 8–15 indicates mild symptoms, 16–25 indicates moderate symptoms, and 26–63 indicates severe symptoms. In a previous study, Cronbach’s α of BAI was 0.92; in addition, BAI was significantly, moderately, and positively correlated with the revised Hamilton Anxiety Scale (*r*_150_ = 0.51) [53]. In this study, Cronbach’s α of BAI was 0.94.

During the measurement of Beck Depression Inventory (BDI), the respondents selected one response for each of the seven items that best represented their feelings in the previous two weeks. The severity of depression was measured on a scale of 0 to 3 points: 0 indicated no depression, and 3 indicated severe depression. A total score of 0–3 indicated minimal depression, 4–6 indicated mild depression, 7–9 indicated moderate depression, and 10–21 indicated severe depression [54]. In a previous study, Cronbach’s α of BDI was 0.86, and its sensitivity and specificity were >82% [55]. In this study, Cronbach’s α of BDI was 0.88.

This study used the 21-item JAREL Spiritual Well-Being Scale developed by Hungelman et al. [56] to assess patients’ spiritual well-being. The items were measured on a six-point Likert scale: 1 indicated strongly disagree, and 6 indicated strongly agree. Seven items were reverse-coded, including the three factors faith/belief dimension, life/self-responsibility, and life satisfaction/self-actualization. The higher the score was, the better the spiritual well-being. Well-being was classified as low (0–50), medium (51–84), or high (85–126). In previous studies, Cronbach’s α of the scale was 0.85 [57,58]; it was 0.84 in this study.

#### 2.4.3. Clinical Characteristics

Patients’ clinical characteristics considered in this study were their perceived health level, comorbidities, HD duration, weekly HD frequency, transportation used to attend HD sessions, accompaniment during HD, and the relationship with the accompanying individual.

#### 2.4.4. Perceptions of and Self-Participation in HD

Suitable instruments for measuring HD patients’ self-participation are lacking. Therefore, to explore well-adjusted patients’ degree of self-participation in HD, we first used qualitative research methods rooted in theory to construct the Patient’s Self-Participation in HD Scale and analyzed it based on qualitative data [59]. Six experts (two researchers, two clinical renal nurses, a head nurse, and the director of the Nephrology Department) were invited to validate the scale’s content. The cultural correlation of each item in the scale was examined; the item content validity index (I-CVI) was 0.92.

The scale included 37 items. Eight items in the first section covered the patients’ perceptions during the previous three months of dialysis. The items were measured on a 5-point Likert scale: 4 indicated always, and 0 indicated never within the last three months. The scale’s Cronbach’s α was 0.83 and consisted of the ambience of the dialysis room (Cronbach’s α = 0.88) and negative emotions during dialysis (Cronbach’s α = 0.93). The second section consisted of 30 items, 29 structured, and 1 open, on patients’ degree of participation in dialysis over the last three months. All items were measured on a 5-point Likert scale: 0 indicated almost impossible (impossible or completion of only 10–20%), and 4 indicated most possible (completion of >90%). The scale’s Cronbach’s α was 0.96. The degree of participation in dialysis covered four dimensions: seven items on creating a new life (Cronbach’s α = 0.93); nine items on implementing self-care (Cronbach’s α = 0.92); nine items on adjusting and living with HD (Cronbach’s α = 0.91); and four items on active sharing and sharing strategies (Cronbach’s α = 0.86).

#### 2.4.5. KDQOL

In 2020, the Research and Development (RAND) Corporation omitted items from and revised the KDQOL-36 scale to produce KDQOL 1.3. Each component summary has a different calculation method based on quality of life over the previous four weeks. The score ranges from 0 to 100 points, with a higher score indicating a better quality of life. Its Cronbach’s α ranges from 0.80 to 0.87 [26,60]. Tao et al. [60] developed the Chinese version of the KDQOL-36 questionnaire, with its Cronbach’s α ranging from 0.69 to 0.78 and a test–retest reliability of >0.70. Cronbach’s α was 0.90 for the overall scale in this study.

### 2.5. Study Procedure

After receiving Institutional Review Board approval, this study recruited participants between 20 January 2020, and 19 April 2021. A medical research assistant trained to administer the questionnaire distributed it to eligible participants who completed it themselves in 20 to 30 min. For those with an education level lower than elementary school, interviews were conducted for 20 to 30 min.

### 2.6. Ethical Considerations

The Institutional Review Board reviewed and approved this study (No. 201900456B0). The participants were aware that they could freely choose to participate in the study and had the right to withdraw from it at any time during the study period. All the questionnaire data were anonymized.

### 2.7. Data Analysis

This study used SPSS 22.0 software (IBM, Armonk, NY, USA) for data analysis. The statistical methods for hypothesis testing included tests for normality, outliers, and multicollinearity. Descriptive statistics are presented as frequency distributions, percentages, and means and standard deviations. Variables with *p* level of <0.25 were analyzed using bivariate linear regression, then individually subjected to a multivariate linear regression of sociodemographic, mental/spiritual, or clinical characteristics. Variables with *p*-level of <0.25 were subjected to a joint multivariate linear regression of sociodemographic, mental/spiritual, or clinical characteristics [61]. The relationships between dependent and independent sociodemographic, psychological and spiritual characteristics, clinical characteristics, and HD self-participation were analyzed using the final multivariate linear regression model.

## 3. Results

### 3.1. Sociodemographic Characteristics

The mean age of patients was 62.24 ± 11.13 years, with a range of 35–91 years. Most patients were male (53.0%), married (74.8%), and had received an elementary school education (31.9%), followed by those who had received a high school (vocational) education (30.2%). Most patients had a religion (83.9%): most were Buddhists (46.0%), followed by Taoists (32.2%). Most patients were unemployed (78.9%) and living with family (98.3%), mainly their spouse (65.8%) or their children (20.8%). The total score range of the HD Social Support Scale was 28 to 72 points, with a mean of 57.92 ± 9.83 points (Table 1).

### 3.2. Psychological and Spiritual Characteristics

The mean total Anxiety Scale score was 10.31 ± 9.73 points, with a range of 0–46 points (Table 2 & Figure 2). The majority of patients (48.7%) had minimal anxiety, followed by 24.2% with mild anxiety, while 27.2% of patients had moderate to severe anxiety. The mean total Depression Scale score was 3.16 ± 4.08 points, with a range of 0–21 points. Most patients (65.1%) had extremely minimal depression, followed by 17.4% with mild depression, and 17.4% with moderate to severe depression. The mean total Spiritual Well-Being Scale score was 87.27 ± 9.98 points, with a range of 63–123 points (Table 2, Table 3 and Table 4 & Figure 2); most participants (53.0%) declared that their spiritual well-being was high.

### 3.3. Clinical Characteristics

The mean time since being diagnosed with an illness requiring HD was 9.38 ± 7.30 years, with a range of 1–33 years (Table 3). Most patients underwent dialysis thrice weekly (97.3%). The patients’ mean self-perceived health status score was 2.19 ± 0.79 points, with most (50.3%) perceiving their status as being normal and 28.9% as being good. Most patients (85.2%) had comorbidities, with hypertension being the most prevalent (65.8%), followed by diabetes (40.6%) and heart disease (24.8%). Most HD patients had one chronic disease (38.9%). Almost half (48.0%) of the patients drove themselves to their dialysis sessions. Most patients were accompanied during their dialysis sessions, mainly by family members (57.7%).

### 3.4. Patient Perceptions of and Self-Participation in HD

The mean total Perception of HD Scale score was 24.19 ± 7.07 points, and the range was 8 to 32 points (Table 4 & Figure 2). The mean Self-Participation in HD Scale score was 78.32 ± 23.15 points, and the range was 12 to 116 points (Table 4 & Figure 2). 

### 3.5. KDQOL

The mean score for patients’ quality of life was 57.46 ± 16.37 points, with a range of 22 to 94 points (Table 4 & Figure 2). The mean total kidney-disease component score (KDCS) was 57.46 ± 16.37 points, and the range was 20 to 100 points. The mean physical component summary (PCS) score was 51.09 ± 20.89 points and the range was 0 to 94 points. The mean total mental component summary (MCS) score was 60.74 ± 22.74 points, and the range was 8 to 100 points.

### 3.6. Factors Related to Quality of Life of HD Patients

The bivariate linear regression analysis indicated that education level (*p* = 0.045), HD social support, degree of anxiety, degree of depression, spiritual well-being status, self-perceived health status, the total number of comorbidities, transportation used to attend HD sessions, accompaniment during HD, HD perception, and HD self-participation were significantly associated with quality of life. However, after controlling for covariates, multivariate linear regression analysis showed that anxiety (*p* < 0.001), self-perceived health status (*p* < 0.001), having two vs. four comorbidities (*p* = 0.042), and self-participation in HD (*p* < 0.001) were the most significant factors associated with quality of life. The overall model was significant (*F*_(24,273)_ = 12.43, *p* < 0.001) and explained 52.2% (*R*^2^ = 0.522) of the variance (adjusted *R*^2^ = 0.480) in quality of life (Table A1).

## 4. Discussion

### 4.1. Psychological and Spiritual Characteristics

This study found that the mean anxiety level of HD patients not infected with COVID-19 was 10.32 ± 9.73, and 51.3% of patients had a BAI score of ≥8.0. This finding is consistent with the results of Schouten et al.’s [62,63] study of HD and peritoneal dialysis patients at 10 dialysis centers in the Netherlands with a mean anxiety level of 10.3 ± 10.1. However, our results were higher than those of Zhang et al. [64], Kurtgoz et al. [65], Zahedian et al. [66], Cukor et al. [46], Wu et al. [67], and Nadort et al. [47]. This inconsistency might be due to different countries, single- or multi-centers, eligible participants, with or without COVID-19 infection, and before or during the COVID-19 pandemic. We suggest that the high anxiety level of patients in our sample is likely due to the poor COVID-19 prognosis reported by the media when the pandemic began [68]. Gedney [33] described the fears of chronic HD patients during COVID-19 from a patient’s and advocate’s perspective: “For those with chronic kidney disease (CKD), on dialysis, or with a transplanted kidney, the world has become terrifying.” The advocate enhanced their knowledge by reading new medical articles published in scientific journals, assisting clinical physicians in disease and treatment management, and delineating the pressing needs of and stress faced by HD patients.

In this study, patients’ mean depression score (BDI-FS) during the pandemic was 3.16 ± 4.08 points, higher than those in Alsaleh et al. [69] and Neitzer et al. [70]. With a BDI-FS cutoff value ≥ 4, the prevalence of depression among the HD patients in this study was 34.8%, higher than those of Neitzer et al. [70], Andrade and Sesso [71], and Alsaleh et al. (27.5%) [69]. However, the prevalence in our study was lower than that in Gao et al. [72], potentially due to different research settings, participants, and the COVID-19 pandemic. As patients must be quarantined during the COVID-19 pandemic, they may develop feelings of loneliness, fear, anxiety, inner guilt, and distress about sequelae (infecting others or family members), which are detrimental to their normal life, social rhythm, and quality of life. The above information suggests that the anxiety and depression felt by HD patients during the COVID-19 pandemic should be emphasized in care provision. The identification of factors that result in depression and anxiety is a key task in nursing care.

The mean score for HD patients’ total spiritual well-being was 87.27 ± 9.98 points, reflecting the high (53.0%) or moderate (47.0%) spiritual well-being of most patients. Currently, no other studies have used the same tools to analyze HD patients’ demographic variables, such as religious beliefs and marital and household status; in our study, 83.9% of the patients had a religious belief, 74.8% were married, 98.3% were living with family members or others, and generally had good social support (57.92 ± 9.83). During their illness, the HD patients came to terms with its personal significance and the limitations and difficulties of overcoming it and achieving spiritual well-being.

### 4.2. KDQOL

This study found that the patients’ mean KDQOL, KDCS, PCS, and MCS scores were 57.46 ± 16.37, 60.54 ± 14.86, 51.09 ± 20.89, and 60.74 ± 22.47 points, respectively. Although the KDQOL score was lower than the reported by Yang et al. [27] and Zhou et al. [24], our findings suggest that the total KDQOL score had improved. However, our score was higher than that of Plantinga et al. [44], Al Wakeel et al. [20], Davison et al. [39], Yang et al. [34], Moura et al. [22], Zimbudzi et al. [51], Pan et al. [29], Vo et al. [43], Cohen et al. [47], Doan et al. [31], Pretto et al. [26], Yazawa et al. [49], De Olivera et al. [40], Legrand et al. [73], Wilkinson et al. [50], and Nadort et al. [47]. This finding may be due to Taiwan having a National Health Insurance scheme that subsidizes patients with major illnesses, injuries, and disabilities. These measures affect patients’ financial status and dialysis compliance, which are associated with healthcare insurance and social welfare, as suggested by previous studies [74,75,76,77,78]. In addition, Taiwanese HD patients’ five-year cumulative survival was 54.2% from 2009 to 2013, higher than that of American patients in 2012 (41.4%) and Canadian patients in 2011 (41.8%; 2020 Taiwan Society of Nephrology Annual Report). This finding shows that Taiwanese HD patients’ KDQOL was still higher than that of other countries despite being determined during the toughest phases of the COVID-19 pandemic.

### 4.3. Factors Related to HD Patients’ Quality of Life

After controlling for covariates, multivariate linear regression analysis showed that Beck Anxiety Inventory score, perceived health status, total comorbidities, and self-participation in hemodialysis were significant factors with a *R*^2^ 0.52 or adjusted *R*^2^ 0.48. Our significant factors and *R*^2^ were different from Garcia-Martnez et al. [30] and Floria et al. [78]. Garcia-Martnez et al. [30] found age and Connor-Davidson Resilience Scale were significant factors for KDQOL total score with a *R*^2^ 0.27, while Floria et al. [78] found age, educational status, and financial status were significant factors for KDQOL total score with an adjusted *R*^2^ 0.28. This might be related to different participants, study framework, and statistical methods. Although *R*^2^ of this study was 0.52, it is higher than Garcia-Martnez et al. [30] and Floria et al. [78]. In addition, Frost [79] indicated *R*^2^ values of the human behavior tended to be less than 50%. The one form of equation of the multivariate linear regression model showed as follows:

KDQOL = 35.02 + (0.42 junior high school vs. ≤elementary school*i*) + (0.05 social support*i*) + (−8.40 BAI score*i*) + (−2.91 BDI score*i*) + (5.12 perceived health status*i*) + (7.52 2 vs. 4 types comorbidity*i*) + (−4.55 3 vs. 2 times of HD per week*i*) + (−1.83 vehicle vs. self-preparation transportation of HD*i*) + (−0.52 no vs. yes accompany during HD*i*) + (0.09 feeling of HD at present*i*) + (0.15 self-participation of HD*i*) = 30.16. Different forms of equation by substituting beta values of categorical variables would have different KDQOL score.

We found that patients with mild, moderate, or severe anxiety had a significantly lower quality of life than those with a minimal level of anxiety. Patients’ KDQOL decreased with increasing anxiety, a finding consistent with that of de Brito et al. [37]. The underlying reason could be that during the COVID-19 pandemic, the Central Epidemic Command Center organized a daily press conference at 2 p.m. to announce in a transparent manner the number of infections and deaths nationwide and the places visited by confirmed cases. The epidemic prevention measures and regulations are also revised on a rolling basis. This information is often headlined by the media, increasing public anxiety. HD patients have to attend sessions two to three times a week and must comply with the rolling epidemic prevention measures and regulations. Going back and forth during dialysis will increase the chance of infection, and they also worry about affecting family members or friends or being stigmatized. Therefore, anxiety or worry increases, which has an impact on their quality of life. HD nurses should adopt diverse nursing strategies and multidisciplinary interventions to query and lessen patients’ anxiety (i.e., by encouraging patients to watch self-preferred television, listen to music for relaxation, and practice mindfulness or meditation in bed while undergoing hemodialysis or staying at home, to improve their self-perceived health, and to be more engaged in their HD life), thereby enhancing their quality of life.

This study found that self-perceived health level is significantly correlated with quality of life. This result differs from that of Yang et al. [34], which may be because they examined the psychological impact of acute public health events on Chinese HD patients and its relationship with their quality of life during the height of the COVID-19 pandemic (March to May 2020) and when it began to wane (December 2020 to January 2021). They discussed the correlations between individual SF-36 and KDQOL scores, unlike this study, which analyzed the correlation based on the total score.

The relationship between quality of life and comorbidities in this study is similar to the findings of several previous studies. HD patients with cardiovascular disease, liver disease, hypertension, hyperlipidemia, and diabetes had a poorer KDQOL [23,26,29,30,32]. This association may be because the renal function of HD patients with these comorbidities could deteriorate if they lacked good illness management, affecting them psychologically, socially, and spiritually, thus reducing their quality of life.

This study found that patients with a higher score for self-participation in HD have a better quality of life; previous studies have shown that patients’ self-efficacy, activation, and self-management skills are positively correlated with quality of life [21,80,81]. Patients’ self-participation in HD is similar to their self-efficacy and self-management, demonstrating the importance of healthcare professionals encouraging patients to be more engaged in their life of HD and monitoring disease status since these variables affect patients’ quality of life. Healthcare teams should thoroughly review the instructional content given to HD patients; this content should include the necessity of self-participation in patient education to understand more about the importance of self-participation in HD to improve patients’ quality of life.

### 4.4. Limitations

A limitation of this study is that its sample consisted only of patients recruited via purposive sampling from the HD center of a single teaching hospital in northern Taiwan, which restricts the generalizability of its results and may cause selection bias. In addition, while the independent variables of this study included the patients’ sociodemographic variables, i.e., mental, spiritual, and clinical characteristics, HD perception, and HD self-participation, laboratory parameters with large variations, Charlson comorbidity index score, HD efficiency, and primary kidney disease were not used as independent variables, which may affect the outcomes.

## 5. Conclusions

During the COVID-19 pandemic, this study found that HD patients with mild, moderate, or severe anxiety had a poor quality of life during the COVID-19 pandemic, whereas those with better self-perceived health, fewer comorbidities, and higher self-participation scores had a better quality of life. Nursing staff and hemodialysis patients have anxious feelings by the rolling revision of the epidemic prevention policy of the epidemic prevention command center during the period of COVID 19, such as the infectivity of COVID-19, the severity of infection after infection, and the stigma environments. Nursing staff are gatekeepers to prevent the spread of the COVID 19 epidemic and are under high pressure and have to ask each hemodialysis patient’s TOCC (Travel history, occupation, contact history, cluster). Nurses are worried about being infected by hemodialysis patients, then infecting their family members. The hurried work steps and nervous tone of nursing staff in the hemodialysis center also make hemodialysis patients more anxious. Thus, the hemodialysis clinic establishes a special area to provide hemodialysis service for hemodialysis patients infected with COVID. Nursing staff also need to be aware of their own anxious feeding, reduce stress through stress self-management strategies, and lower the volume of speaking. While patients have high anxiety, they need to accompany and comfort them. Emotional care can be provided by nursing staff who are more trusted by hemodialysis patients, providing a familiar, safe, and comfortable environment and an opportunity to discuss their inner fears. Furthermore, the individual differences in age, education level, and anxiety level of hemodialysis patients should be considered when implementing nursing guidance, so as to improve the quality of hemodialysis care for hemodialysis patients. Future studies should compare the quality of life of patients from the same cohort before and after the COVID-19 pandemic.

## Figures and Tables

**Figure 1 healthcare-11-01155-f001:**
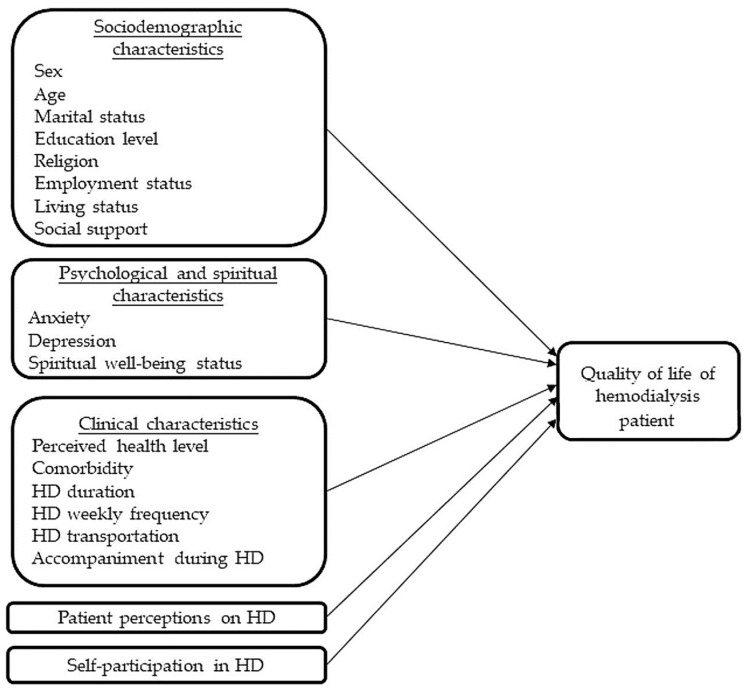
The study framework.

**Figure 2 healthcare-11-01155-f002:**
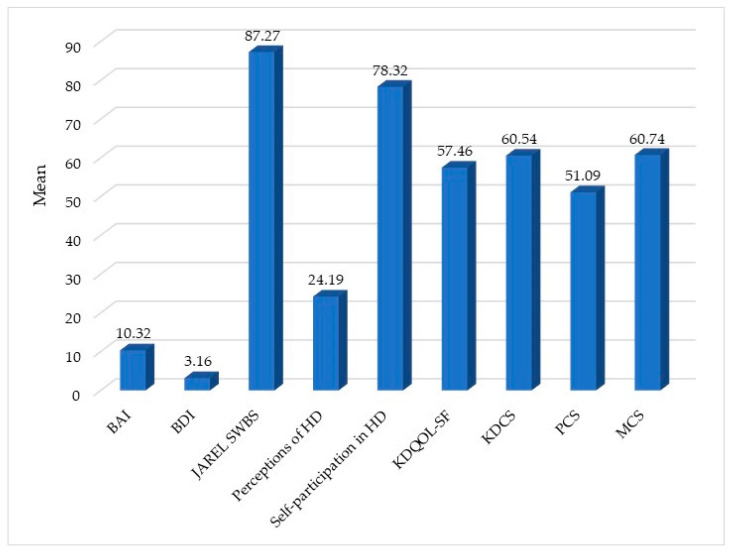
Means of variables.

**Table 1 healthcare-11-01155-t001:** Sociodemographics of hemodialysis patients (*n* = 298).

Variable	Category	Range	Mean (SD)	*n* (%)
Age in years		35–91	62.24 (11.13)	
Sex	Female			140 (47.0)
	Male			158 (53.0)
Marital status	Single			28 (9.4)
	Married			223 (74.8)
	Divorced/living together			17 (5.7)
	Widow			30 (10.1)
Education level	≤Elementary			95 (31.9)
	Junior high school			66 (22.1)
	High school/vocational school			90 (30.2)
	Diploma (2-year, 3-year, or 5-year program)			31 (10.4)
	≥University			16 (5.4)
Religion	No			48 (16.1)
	Yes			250 (83.9)
	Buddhist			137 (46.0)
	Taoist			96 (32.2)
	Christian/Catholic			13 (4.4)
	I-Kuan Tao			4 (1.3)
Employment status	No			235 (78.9)
	Yes			63 (21.1)
	Part-time			20 (6.7)
	Full-time			43 (14.4)
Living status	Living alone			5 (1.7)
	Living with relative or other			293 (98.3)
	Spouse			196 (65.8)
	Children			62 (20.8)
	Relatives			24 (8.1)
	Friend/nursing aids			5 (1.7)
	Other			6 (2.0)
Social Support		28–72	57.92 (9.83)	
	Family, relatives, and friends’ dimension	16–36	29.11 (4.64)	
	Medical and nursing staff dimension	11–36	28.81 (6.29)	

**Table 2 healthcare-11-01155-t002:** Mental and spiritual characteristics of hemodialysis patients (*n* = 298).

Variable	Items	Range	Mean (SD)	*n* (%)
Beck Anxiety Inventory	21	0–46	10.32 (9.73)	
0–7 (Minimal level of anxiety)				145 (48.7)
8–15 (Mild anxiety)				72 (24.2)
16–25 (Moderate anxiety)				55 (18.5)
26–63 (Severe anxiety)				26 (8.7)
Beck Depression Inventory	7	0–21	3.16 (4.08)	
0–3 (Minimal depression)				194 (65.1)
4–6 (Mild depression)				52 (17.4)
7–9 (Moderate depression)				18 (6.0)
10–21 (Severe depression)				34 (11.4)
JAREL Spiritual Well-Being Scale	21	63–123	87.27 (9.98)	
51–84 (Moderate)				140 (47.0)
85–126 (High)				158 (53.0)

**Table 3 healthcare-11-01155-t003:** Clinical characteristics of hemodialysis patients (*n* = 298).

Variable	Category	Range	Mean (SD)	*n* (%)
Perceived health level	Very poor	0–4	2.19 (0.79)	4 (1.3)
	Poor			46 (15.4)
	Common			150 (50.3)
	Good			86 (28.9)
	Very good			12 (4.0)
Comorbidity	No	0–4	1.53 (1.03)	44 (14.8)
	Yes (multiple choice)			254(85.2)
	Hypertension			196 (65.8)
	Diabetes			121 (40.6)
	Heart disease			74 (24.8)
	Arthritis			21 (7.0)
	Stroke			22 (7.4)
	Other			22 (7.4)
Number of comorbidities	None	0–4	1.53 (1.03)	44 (14.8)
	1 type			116 (38.9)
	2 types			87 (29.2)
	3 types			38 (12.8)
	4 types			13 (4.4)
HD duration in years		0–33	9.38 (7.30)	
Frequency of hemodialysis per week	3 times			290 (97.3)
	2 times			8 (2.7)
HD transportation	Self-preparation (taxi or Rehabus)			57 (19.1)
	Vehicle			143 (48.0)
	Motorcycle			24 (8.1)
	Bus/transit/Formosa Fairway Corporation			54 (18.1)
	Other			20 (6.7)
Accompaniment during HD	No			114 (38.3)
	Yes			184 (61.7)
	Family member			172 (57.7)
	Friend/relative			5 (1.7)
	Nursing aid			7 (2.3)

**Table 4 healthcare-11-01155-t004:** Perceived feelings about hemodialysis, self-participation, and quality of life of hemodialysis patients (n = 298).

Variable	Item	Range	Mean (SD)
Perceptions of HD	8	8–32	24.19 (7.07)
Self-participation in HD	29	12–116	78.32 (23.15)
Kidney Disease Quality of Life Short Form (KDQOL-SF^tm^)	40	22–94	57.46 (16.37)
Kidney disease composite summary (KDCS)	24	20–100	60.54 (14.86)
Physical composite summary (PCS)	9	0–94	51.09 (20.89)
Mental composite summary (MCS)	7	8–100	60.74 (22.47)

## Data Availability

The data used in this study are not available to the public due to ethical considerations.

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
