# Peer review of "Factors Related to Quality of Life of Hemodialysis Patients during the COVID-19 Pandemic"

_healthcare, 2023, doi:10.3390/healthcare11081155_

Round 1

Reviewer 1 Report

In this manuscript, Szu et al. analyzed factors associated with quality of life of hemodialysis patients during the COVID-19 pandemic. They collected data including patients' sociodemographic, psychological, and clinical characteristics and found that anxiety, self-perceived health status, comorbidities, and self-participation in hemodialysis were significantly associated with quality of life. These findings can have important implications to help improve nursing strategies for hemodialysis patients. The manuscript is well-written, and the conclusions are supported by the data. I only have minor comments:

- In the introduction, it would be better to add a brief introduction to COVID-19, and how it might affect hemodialysis patients.

- In line 284, the word "eligiable" should be "eligible".

Author Response

Point 1: In the introduction, it would be better to add a brief introduction to COVID-19, and how it might affect hemodialysis patients.

Response 1: Three sentences have been added to emphasize how COVID-19 might affect hemodialysis patients. (p.2, Line 75-82)

Point 2In line 284, the word "eligiable" should be "eligible".

Response 2: The word "eligiable" has been revised as "eligible". (p.10, Line 298)

Reviewer 2 Report

The paper is well organized and the topic is attractive for the scientific community.  I suggest this paper for publication. For a  minor point, the data for the statistical analysis can be represented with a color scale in a graph.

1) R2 for the model (0.552) is quite low. Please discuss the possible enhancement in the model to increase the R2. Please provide the open forms of the equations that you used and explain each term there.

2) Please compare your model with the literature (considering other international hemodialysis reports after pandemic)

3) Can you also prepare a similarity data analysis for your data and the published work -if the data is available of course-? So, the same trends can be discussed. 

4) For the data analysis part, please prepare a figure showing the major categories with the mean values  based on the number of patients.

Author Response

Point 1: The paper is well organized and the topic is attractive for the scientific community. I suggest this paper for publication. For a minor point, the data for the statistical analysis can be represented with a color scale in a graph.

Response 1: A figure of showing the major categories with the mean values based on the number of patients has been added (p.9, Line 277-278).

Point 2: R2 for the model (0.552) is quite low. Please discuss the possible enhancement in the model to increase the R2. Please provide the open forms of the equations that you used and explain each term there.

Response 2: Frost, J. (n.d.) indicates “R-squared does not indicate if a regression model provides an adequate fit to your data. A good model can have a low R2 value. On the other hand, a biased model can have a high R2 value! R-squared should accurately reflect the percentage of the dependent variable variation that the linear model explains. Your R2 should not be any higher or lower than this value. The correct R2 value depends on your study area. Different research questions have different amounts of variability that are inherently unexplainable. Case in point, humans are hard to predict. Any study that attempts to predict human behavior will tend to have R-squared values less than 50%. However, the interpretation of the significant relationships in a regression model does not change regardless of whether your R2 is 15% or 85%! The regression coefficients define the relationship between each independent variable and the dependent variable. The interpretation of the coefficients doesn’t change based on the value of R-squared. An unbiased model has residuals that are randomly scattered around zero. Non-random residual patterns indicate a bad fit despite a high R2. Always check your residual plots!” Residuals of our model are randomly scattered around zero. Standardized residual = -2.28 – 2.20 and Cook distance = 0.000 – 0.043. The section has been revised (p.11, 354-355; p.11, Line 356-363).

References:

Frost, J. (n.d.). How to interpret R-squared in regression analysis. Retrieved from https://statisticsbyjim.com/regression/interpret-r-squared-regression/

Frost, J. (n.d.). How High Does R-squared Need to Be?. Retrieved from https://statisticsbyjim.com/regression/how-high-r-squared/

Frost, J. (2019). Regression analysis: An intuitive guide for using and interpreting linear models. Author.

Point 3: Please compare your model with the literature (considering other international hemodialysis reports after pandemic)

Response 3: We search literature again. However, there are no multivariate linear regression model using same instrument of KDQOL after pandemic. We compared our model with the literature, which conducted survey near the end of 2019 (pp.11, Line 348-354).

Point 4: Can you also prepare a similarity data analysis for your data and the published work -if the data is available of course-? So, the same trends can be discussed.

Response 4: Thanks for suggesting trend analysis by analyzing our data with the published work. However, the data is not available.

Point 5: For the data analysis part, please prepare a figure showing the major categories with the mean values based on the number of patients.

Response 5: A figure of showing the major categories with the mean values based on the number of patients has been added (p.9, Line 277-278).

Reviewer 3 Report

Szu et al. conducted a questionnaire-based study titled “Factors Related to Quality of Life of Haemodialysis Patients during the COVID-19 Pandemic”. The authors present a manuscript about factors related to haemodialysis patients’ quality of life during the COVID-19 pandemic. Although the manuscript is of interest, my main concerns are as follows:

-       A sample size calculation was conducted and suggested 219 patients, however, a total of 298 were included in the final analysis. In addition, the authors state that their recruitment method included purposive sampling, which implies the “intentional selection of informants based on their ability to elucidate a specific theme, concept, or phenomenon.“  (DOI: 10.1007/978-94-007-0753-5_2337). The authors do not state, how many, or which of the total number of outpatients were offered the possibility to participate in the trial. This introduces a severe selection bias, as it is unclear how these participants were chosen.

-       The present study does not include a control group. Hence, it is not possible to conclude that these results are specific for HD in the COVID period. 

-       Conclusions in lines 377-385 are not supported by the present data. Unless already performed, comparing the quality of life of the same cohort before vs. after the COVID pandemic will not be possible in future studies.

Author Response

Point 1: A sample size calculation was conducted and suggested 219 patients, however, a total of 298 were included in the final analysis. In addition, the authors state that their recruitment method included purposive sampling, which implies the “intentional selection of informants based on their ability to elucidate a specific theme, concept, or phenomenon. (DOI: 10.1007/978-94-007-0753-5_2337). The authors do not state, how many, or which of the total number of outpatients were offered the possibility to participate in the trial. This introduces a severe selection bias, as it is unclear how these participants were chosen.

Response 1: Concerning the valid questionnaires and factor analysis for understanding construct validity of the new development scale (perceptions of and self-participation), the actual sample size was larger than the estimated sample size. This study recruited participants from the target population about 750 using purposive sampling. Although the purposive sampling has its weakness, probability sampling methods were not feasible for this study. Therefore, this weakness has been addressed in the limitations (p.11, Line 395-400). This comment has been revised (p.3 Line 113-114; p.4, Line 124-126; p.12, Line 400).

Point 2: -The present study does not include a control group. Hence, it is not possible to conclude that these results are specific for HD in the COVID period.

Response 2: The first sentence of the conclusion has been revised (p.12, Line 407).

Point 3: -Conclusions in lines 377-385 are not supported by the present data. Unless already performed, comparing the quality of life of the same cohort before vs. after the COVID pandemic will not be possible in future studies.

Response 3: The first sentence of the conclusion has been revised (p.12, Line 407).

Round 2

Reviewer 1 Report

The manuscript is finely revised and sufficiently improved by the authors. I have no more concerns.

Reviewer 2 Report

It is now suitable for publication.

Reviewer 3 Report

Point 3 was not adequately addressed.

Claims in lines 410-490 are not justified by the present data and should be removed from the conclusions section.

The authors may consider discussing these claims in the discussion section of the manuscript.

Author Response

Point 1: Point 3 was not adequately addressed.

Response 1: This section has been revised again (p.12, Line 407-428).

Point 2: Claims in lines 410-419 are not justified by the present data and should be removed from the conclusions section. The authors may consider discussing these claims in the discussion section of the manuscript.

Response 2: This section has been revised again (p.11, Line 376-381; p.12, Line 401-404; p.12, Line 415-437).

Round 3

Reviewer 3 Report

I have no further comments.